



# Investigating the impact of regional transport on PM2.5 formation using vertical observation during APEC 2014 Summit in Beijing

Yang Hua[1,2], Shuxiao Wang[1,2*], Jiandong Wang[1,2], Jingkun Jiang[1,2], Tianshu Zhang[3], Yu Song[4], Ling Kang[4], Wei Zhou[1,2], Runlong Cai[1,2], Di Wu[1,2], Siwei Fan[1,2], Tong Wang[1,2], Xiaoqing Tang[5], Qiang Wei[6], Feng Sun[6], Zhimei Xiao[7]

[1]State Key Joint Laboratory of Environment Simulation and Pollution Control, School of Environment, Tsinghua University, Beijing 100084, China.

[2]State Environmental Protection Key Laboratory of Sources and Control of Air Pollution Complex, Beijing 100084, China

[3]Anhui Institute of Optics and Fine Mechanics, Chinese Academy of Sciences, Hefei 230031, China

[4]College of Environmental Sciences and Engineering, Peking University, Beijing, 100871, China

[5] Hebei Environmental Monitoring Center, Hebei 050051, China

[6] Beijing Environmental Monitoring Center, Beijing 100048, China

[7] Tianjin Environmental Monitoring Center, Tianjin 300191, China

Corresponding to: Shuxiao Wang (shxwang@tsinghua.edu.cn)

**ABSTRACT**

During APEC (Asia-Pacific Economic Cooperation) Economic Leaders' 2014 Summit in Beijing, strict regional air emission control was implemented, providing a unique opportunity to investigate the transport and formation mechanism of fine particulate matter ($PM_{2.5}$). This study explores the use of vertical observation methods to investigate the influence of regional transport on $PM_{2.5}$ pollution in Beijing before and during the APEC Summit. Vertical profiles of extinction coefficient, wind, temperature and relative humidity were monitored. Three $PM_{2.5}$ pollution episodes were analysed. In episode 1 (October 27th to November 1st), regional transport accompanied with the accumulation of pollutants under unfavourable meteorological conditions led to the pollution. In episode 2 (November 2nd to 5th), pollutants left from episode 1 were retained in the boundary layer for 2 days in the region and then settled down to the surface, leading to an explosive increase of $PM_{2.5}$. The regional transport of aged aerosols played a crucial role in the heavy $PM_{2.5}$ pollution. In episode 3 (November 6th to 11th), emission from large point sources had been controlled for several days while primary emissions from diesel vehicle might lead to the pollution. It is found that ground-level observation of meteorology condition and air quality could not fully explain the pollution process while vertical parameters





(aerosol optical profile, wind profile, relatively humidity profile and temperature profile) improved
the understanding of regional transport influence on heavy pollution process. Further vertical
observations are needed to investigate the pollutants transport especially during the explosive increase
pollution episode.





**1. Introduction**
With a rapid economic development and increases in energy consumption, large quantity of emissions
has caused serious air pollution in China. Monitoring data show that Beijing-Tianjin-Hebei (BTH)
region is one of the most polluted region in China (Zhao et al., 2013; Wang et al., 2014). The region
was home to eight out of the top 10 most polluted Chinese cities in 2014 (MEP-Ministry of
Environment Protection, 2015). In 2014, the annual average $PM_{2.5}$ (particulate matter with
aerodynamic diameter less than 2.5 μm) concentration reached 95 μg/m$^3$ in the BTH region. With
21.5 million residents and 5.3 million vehicles, Beijing has been burdened with severe pollution
episodes frequently in recent years (Beijing Municipal Bureau of Statistics, 2014). The capital is
surrounded by mountains in three directions (north, west and east). The top three most polluted cities
in China (Baoding, Xingtai and Shijiazhuang) are located in the south to Beijing. Polluted air mass
from the south contributes to $PM_{2.5}$ pollution in Beijing (Wang et al., 2015). Source apportionment by
Beijing Environmental Protection Bureau indicates regional transport contributed 28%-36% to $PM_{2.5}$
in Beijing in 2012-2013. During some severe pollution periods, regional contribution was more than
50% (http://www.bjepb.gov.cn/bjepb/413526/331443/331937/333896/396191/index.html). Quite a
few researches have studied the causes of heavy polluted episodes in BTH region and show regional
transport plays an important role in pollution formation. The sharp $PM_{2.5}$ build-up events in Beijing
were unique while accumulation pollution process occurred at other cities in the region. This
indicated that $PM_{2.5}$ was probably transported to Beijing from other cities (Zheng et al., 2015; Ji et al.,
2014; Tao et al, 2014; Zhao et al., 2013). In the meanwhile, most severe pollutions are under stable
synoptic meteorological conditions in Beijing (Sun et al., 2015; Zheng et al., 2015; Zhao et al., 2013).
The low wind speed and stable synoptic meteorological condition at ground level cannot explain the
reason that regional transport makes significant contribution to severe pollution. A previous study has
shown the secondary aerosol in Beijing probably mainly formed over regional transport according to a
vertical observation from the ground to 260m height. (Sun et al., 2015). Therefore, vertical profiles of
meteorology and air quality might help us to understand the impacts of regional transport to heavy
pollution during stagnant conditions.
As in other megacities with local sources and regional transport, air quality in Beijing are affected by
many factors, including emissions inside the city, formation of secondary pollutants, atmospheric
mixing, and regional transport. It has been well known that the strength of each factor varies
according to emissions and/or weather conditions. Therefore, it is challenging to pin point the major
contributors in any given time periods, either clean or polluted episodes. This is especially difficult in
BTH region considering the complicated emission sources and transport processes.



Emission control measures implemented during some events provide a unique opportunity to
investigate the impact of various factors influencing air quality. One of them was APEC (Asia-Pacific
Economic Cooperation) Economic Leaders' 2014 Summit held in Beijing from November 5[th] to 11[th],
2014. A strict air pollution control plan was carried out in the BTH Region to improve air quality in
Beijing from November 2[nd] to 11[th] for APEC. According to a conservative estimate by MEP,
production of 9,289 plants were paused and 3900 plants were running at reduced capacity in six
provinces (Beijing, Tianjin, Hebei, Shanxi, Shandong and Inner Mongolia); and more than 40
thousand          construction          sites          were          shut          down          temporally
(http://www.zhb.gov.cn/gkml/hbb/qt/201411/t20141115_291482.htm). Other measures include traffic
control (50% of private passenger vehicles and 70% of buses were off-road) and frequent road
sweeping and cleaning in Beijing. More detail emission control measures are supplied in the
supporting information. Studies have found that regional emission control effectively reduced air
pollutant concentrations during the Summit (Wen et al., 2015; Tang et al., 2015; Han et al., 2015;
Chen et al., 2015; Sun et al., 2016a). The significantly reduced local emissions led to reduced
complexity of pollution process, thus providing a unique opportunity to investigate the influence of
transport events on $PM_{2.5}$ levels in Beijing.
The objective of the study is to investigate the impact of regional transport on $PM_{2.5}$ in Beijing using
both ground-level and vertical observations. Field observation was conducted at a rural site (Liulihe)
in southwest Beijing before and during the control period of the APEC 2014 Summit. Vertical profiles
of temperature, RH (relative humidity), wind speed and direction, and extinction coefficient were
observed as well as pollutants concentration and meteorological parameters on the ground. The
characteristics of three $PM_{2.5}$ pollution episodes were analysed. Findings of this study will help
explore vertical observation methods for in-depth analysis of the meteorological and transport
influence. Furthermore, it can aid the development of future air quality management strategies in BTH
and other regions around the globe, including emission control and air surveillance.
**2. Field observation and analysis methods**
**2.1 Field observation site and sampling methods**
Beijing is surrounded by mountains in the west, north and east directions, which blocks the pollutants
from spreading. The open air corridor in the south exposes the capital to air mass passing Hebei
Province (Fig. S1) a heavily polluted area in China. To investigate the impact of regional transport on
Beijing, a rural site (Liulihe site, 116°2′E, 39°36′N) was chosen in the southwest of Beijing. It was
located on the border of Beijing and Hebei Province (Fig. S1).





The field campaign was conducted from October 27th to November 12th, 2014, including both ground-
level and vertical observations. Detailed information of instruments at Liulihe site is provided in
Table S1. Ground-level observations included meteorological parameters, mass concentration of
$PM_{2.5}/PM_{10}$, $SO_2$, $NO_x$ and $O_3$ as well as physical and chemistry properties of PM. $PM_{2.5}/PM_{10}$ mass
concentration was determined by the TEOM method. Particle size distribution from 3nm to 10μm
were measured by a spectrometer assembled in-house including one Nano scanning mobility particle
sizers (NSMPS), one scanning mobility particle sizers (SMPS), and one aerodynamic particle sizer
(APS) (Liu et al., 2014).
ACSM (Aerosol Chemical Speciation Monitor), a low-maintenance aerosol mass spectrometer, was
used to measure non-refractory (NR) particulate matter with aerodynamic diameters smaller than 1μm
($PM_1$) (Ng et al., 2011). The ACSM data was calibrated with a collection efficiency (CE) value to
compensate for the particle loss. The CE value of 0.45 recommended by Middlebrook et al. (2012)
based on the monitoring site condition (see supporting information) was used in this study. The NR-
$PM_1$ concentration measured by ACSM tracks well with $PM_{2.5}$ measured by the TEOM ($R^2$=0.91) and
the regression slope is 0.43 (Fig. S2). Positive matrix factorization (PMF) with the PMF2.exe
algorithm was used to distinguish different components of OA measured by ACSM (Paatero and
Tapper, 1994). The PMF was performed and evaluated following the PMF analysis guide
(http://cires1.colorado.edu/jimenez-group/wiki/index.php/PMF-AMS_Analysis_Guide). Three factors
were distinguished (Fig. S3), i.e., HOA (hydrocarbon-like organic aerosol), SVOOA (semi volatility
oxygenated organic aerosol) and LVOOA (low volatility oxygenated organic aerosol).
Beyond ground-level concentrations of routinely monitored air pollutants and meteorological
parameters, the assessment was aided by vertical observations including vertical extinction coefficient
profile, as well as vertical wind, RH and temperature profiles. The vertical extinction coefficient
profiles depict the distribution of PM, which could be used to infer mixing process of particles
transported in from high evaluations and those near the ground. Vertical wind profile can help figure
out the transport direction. Vertical RH profile can provide the RH information at transport layers,
thus helping investigate heterogeneous reaction at the layers. Vertical temperature profiles provide
information on the stability of and mixing in the boundary layer. Lidar was used to observe the
vertical optical properties of atmospheric aerosols at Liulihe site. The lidar consists of three parts,
including emitting system, receiving system and signal analogue system (Chen et al., 2015). The laser
source emitted pulse at 355/532nm. The pulse energy is 30MJ at 355nm and 20MJ at 532nm. The
pulse repetition is 20Hz. The telescope for receiving system is based on a Cassegrain design.
Diameter of the telescope is 200mm with a vertical resolution of 7.5m. The particle backscatter





coefficient and extinction coefficient was retrieved by Fernald method (Fernald et al., 1984). CFL-03
phased array wind profile radar was used to monitor the vertical wind speed and direction with
resolutions of 50 m (0-1 km) and 100 m (1-5.5 km). Parameters of these instruments can be found in
another paper (Wang et al., 2013). Vertical profiles of atmospheric temperature and humidity were
derived by profiling radiometers. The channel centre frequencies were 22-32 GHz (K-Band) and 51-
59 GHz (V-Band). The vertical resolutions were 60 m (0-4 km) and 120 m (4-10 km).
**2.2 Back trajectory analysis**
Trajstat, a GIS-based software into which the HYSPLIT (Hybrid Single Particle Lagrangian
Integrated Trajectory) model was loaded (Wang et al., 2009), was used to calculate the back trajectory.
The model was run every 6 hours in a 24-hour mode back-trajectory mode at 1000 m above sea level
from Liulihe site to identify the origins and path way of air mass. The meteorology data used in the
mode was obtained from the Global Data Assimilation System (GDAS) model
(http://www.ready.noaa.gov/READYamet.php).
**2.3 Quantification of regional transport contribution**
A novel technique was used to quantify the contribution of regional transport. The diurnal trend of
$PM_{2.5}$ in Beijing often exhibit "Saw-tooth cycles" with a smoothly increasing or decreasing baseline
upon which daily cycles are superimposed. Ancillary measurements around Beijing show that the
baselines represent regional aerosols, while the daily cycles represent local aerosols. Following Jia et
al. (2008), the total contribution is defined as the area under the concentration line ($A_t$), while its
regional component is defined as the area under the baseline curve ($A_r$). Both areas are approximated
using trapezoid numerical integration as Eq. (1):
$$A_N = \sum_{n=1}^{N-1} A_i = \sum_{n=1}^{N-1} \frac{(C_i + C_{i+1})}{2} \times (t_{i+1} - t_i) \tag{1}$$
Where N is the total number of hourly $PM_{2.5}$ concentrations in a specific time period, $C_i$ is total
concentration (for $A_t$) or baseline concentration (for $A_r$) value at time $t_i$ (i=1, N-1). The baseline
concentration curve is the line connecting daily afternoon minimal values. The percentage regional
contribution ($R$) is expressed as following Eq. (2):
$$R = \frac{A_r}{A_t} \times 100\% \tag{2}$$





**3. Results and discussion**
**3.1 General characteristics of atmospheric pollution before and during APEC summit**
To investigate the changes in air quality during APEC summit, average pollutant concentrations and
the rates of changes were calculated. Period 1 (October 27th to November 2nd) and period 2
(November 3rd to November 12th) were defined to represent the periods before and during the APEC
summit. Concentrations of $PM_{2.5}$, $SO_2$ and $NO_2$ decreased significantly during the emission control
(Period 2) compared to before control (Period 1) as shown in Fig. S4 (a). The large rates of reduction
were observed for $NO_2$ (37%) and $SO_2$ (36%), while the reduction in $PM_{2.5}$ was smaller (21%) but still
significant (Fig. S4 (b)).
Three pollution episodes were selected to discuss the pollution characteristics during the observation
(Fig. S5). $PM_{2.5}$ concentration at Miyun site (locate in northern Beijing, shown in Fig. S1, data source:
Beijing EPB) is shown in Fig. S5 alongside Luilihe to demonstrate the synchronism of $PM_{2.5}$ levels at
different sides in Beijing. Episode 1 (October 27th to November 1st) represents the period before the
emission control. Episode 2 (November 2nd to 5th) was the first pollution episode during the emission
control plan. Episode 3 (November 6th to 11th) was the second pollution episode during the emission
control plan. At Luilihe, $PM_{2.5}$ concentration was the highest in episode 1 ($140\pm70\mu g/m^3$) before
implementation of emission control, whereas the mean values were close during the last two episodes
($91\pm75\mu g/m^3$ and $89\pm61\mu g/m^3$).
The average concentration of online $PM_1$ chemical components was shown in Fig. 1. Average
concentrations of OM (organic matter), $NH_4^+$, $SO_4^{2-}$ and $NO_3^-$ were the highest in episode 1 before
emission control. During episode 2, those compounds decreased by 32-60%. In episode 3, the average
concentrations remained similar except $NH_4^+$ which decreased by 12%. HOA (related to primary
emission), LVOOA and SVOOA were distinguished. Compared with episode1, the HOA, LVOOA
and SVOOA decreased by 22%, 58% and 28% in episode 2. After that, LVOOA kept decreasing by
10% in episode 3 while HOA and SVOOA increased by 39% and 5%.
Overall, most meteorological parameters changed little during the three episodes except RH (Fig. S6)).
The average ground-level RH (69%) in episode 1 was higher compared with those in episode 2 (50%)
and in episode 3 (58%). Wind speed remained low during the entire observation. The average wind
speed was 0.5m/s, 0.8m/s and 0.7m/s in episode 1, episode 2 and episode 3, respectively. The
dominant wind direction was southwest during the 10-day observation. The frequency of southwest



wind was above 60% during each of the three episodes, with the highest occurrence of 81% observed
during episode 2.
The significant reduction in pollutant concentrations during APEC shown above implied that the
emission control was effective. However, the general characteristics derived from ground-level
observation are insufficient to identify the leading cause of air pollution, local emissions, regional
transport, or both. Furthermore, the significant differences of particle chemical components changes
from episode 2 to episode 3 under similar ground-level meteorological conditions and local emission
intensity suggest different transport or formation mechanisms during those two episodes. Therefore,
vertical observations will be used to aid further investigation in each of the three episodes in the
following section.
**3.2 Characteristics of heavy PM$_{2.5}$ pollution episodes and contribution of regional transport**
**3.2.1 Pollution process in episode 1**
Episode 1 (October 27th to November 1st) was before emission control. The high level of PM$_{2.5}$ is
typical in Beijing during the autumn. There were two unique features in this episode. One is the
continued increases of PM$_{2.5}$ mass and PM$_1$ component concentrations during the first four days, with
OM showing a more distinct diurnal cycle (Fig. 2, Fig. 3 and Fig. S5). Another is the rapid increase of
OM on Oct 29th (Fig. 3). Both suggest except secondary formation, other mechanisms might impact
the OM growth and needs further investigation.
Various parameters collected during episode 1 are shown in Fig. 4. Combining the ground-level
observation and vertical observation, it is evidenced that the pollution was caused by the regional
transport and pollutants accumulation later. Vertical extinction coefficient data observed at
Yongledian site (116°47′E, 39°43′N) near Liulihe site were used (Fig. 4(a)), because the optical lidar
at Liulihe didn't work in October. High level of PM appeared at approximately 2 km above ground
(Fig. 4 (a)) and retained there for 1 day. The air mass came from the southwest where emissions were
high (see horizontal wind direction profile, Fig. 4 (c)). Back trajectories also show air mass from
southwest arrived in Liulihe, as well as Yongledian (Fig. S7). Then pollutants settled down (see
downward vertical wind direction in Fig. 4 (b)) and mixed with aerosols on the ground (Fig. 4 (a)).
The online particle size distribution also implied the transport process. During the same period (from
13:00 to 20:00 on October 28th), a new group of particles appeared and mixed with existing particles,
indicating the arrival of aged aerosols (Fig. 4 (e)). As mentioned above, except secondary formation,
other mechanisms might impact OM increase. The increase of OM might come from freshly-emitted





organic particles and transported to the site instead of aged particles. One evidence is that both HOA
and OOA increased significantly. Another is that the OM peak appeared after the transport occurrence,
much earlier than SNA. It is noticed, even wind direction on the ground changed to north in the early
morning on October 29th, it still kept in the southwest above 500m, indicating significant influence of
regional transport.
In the next two days (October 30th to 31st), vertical wind direction was downward and pollutants were
easily accumulated in the boundary layer (Fig. 5). Meanwhile, high RH on the surface (Fig. S6)
enhanced the formation of SA (secondary aerosol) as pointed out by Pathak et al. (2009). Under this
condition, $NH_4^+$, $SO_4^{2-}$ and $NO_3^-$ concentrations increased at rates of 0.26μg/m$^3$/h, 0.21μg/m$^3$/h, and
0.58μg/m$^3$/h, respectively. The peak of $NH_4^+$, $SO_4^{2-}$ and $NO_3^-$ concentrations was two days later than
OM. This also proved the organic particles were transported to Beijing and reached to the peak on
October 29th and secondary formation became severe later, both of which promoted the pollution
occurrence.
To quantify the impacts of regional transport, the transport component is calculated with the method
introduced in section 2.2. The baseline needs to be defined first especially for pollution end timing.
Here the vertical observation and ground observation were combined to discuss when the pollution
ended. In the morning on 1st November, air mass from the north above 1000 m arrived Beijing. The
vertical temperature gradient decreased and vertical mixing became weak (wind vertical speed was
very low). Consequently, PM$_{2.5}$ accumulated and had a sharp increase. Then clean and cold wind from
north caused sharp increase of wind speed and decrease of atmosphere pressure. Based on the analysis
above, pollution ended up at 18:00 when the week temperature ended and PM$_{2.5}$ decreased sharply
(Fig. 6). The regional component is calculated based on the determination of baseline.
For episode 1, the regional component accounted for 75%, indicating the important influence of
regional transport on the pollution. It can be seen that episode 1 was a pollution episode influenced by
transport process in Beijing. RH was high, wind speed kept low and wind direction was dominated by
southwest in the surface. Vertical observation showed pollutants transported from southwest settled
down. OM concentration increased significantly when the transport PM was observed. After that
vertical wind direction kept downward and promoted the pollutants accumulation, especially SNA.
**3.2.2 Pollution process in episode 2**
Episode 2 (November 2nd to 5th) saw a lower mean PM$_{2.5}$ concentration (91±75μg/m$^3$) due to the
implementation of emission control since November 2nd. Unlike the gradual accumulation of PM





observed in episode 1, $PM_{2.5}$, OM and SNA had a sharp increase from November 4th to 5th. The
concentrations of $NH_4^+$, $SO_4^{2-}$ and $NO_3^-$ increased at rates from the lowest to the highest of
0.88μg/m³/h, 0.43μg/m³/h, and 1.64μg/m³/h, respectively, much faster than that in episode 1. OOA
also increased much more significantly during this episode. The explosively increases of PM
components mainly SA in such a short period of time is contrary to lower RH values in this episode
leading to less heterogeneous reaction. Thus, such rapid increases in PM levels could be transport of
aged aerosol from other regions, as hypothesized by previous studies where the transport process
wasn't observed directly (Yue, et al., 2009; Massling., et al, 2009; Sun et al., 2014; Sun et al., 2016b).
With the aid of vertical observation, an in-depth investigation revealed atmospheric processes leading
to the peak concentrations during November 4th to 5th. Firstly, after the end of episode 1 at November
1st, relatively high PM levels still resided at 1000m (from November 2nd to 3rd) as shown in the
vertical extinction coefficient (Fig. 7). Furthermore, a band of high PM centered around 750 m were
observed ((Fig. S8) on November 3rd at another site (Baoding site, 115°31′E, 38°52′N, shown in Fig.
S1) in the BTH region, suggesting a wide-spread PM aloft in the region. During the next two days, the
pollutants were transported in the region and the slow winds (average speed of 4.8m/s at 1000 m)
allowed aerosols ample time to age in their journey. Back trajectories showed transport of air mass
from the southwest at the night of November 3rd (Fig. S9), consistent with the vertical wind profile
observed at Liulihe (Fig. 8 and Fig.9). On November 4th, the downward motion of air mass around
1000 m above ground intensified, bringing the aged aerosols down and mixing them with the aerosols
on the ground. The well mixed boundary layer with regard to aerosol is evidence in Fig. 9 with a
fairly uniform distribution from the ground to 900 m. Consequently, secondary chemical component
concentrations of $PM_1$ (Fig. 2 and Fig. 3) started ascending with remarkably fast rates.
Dry and clean wind from north direction arrived in the early morning on November 5th. RH started to
increase significantly at 10:00 and wind speed became higher from 12:00. At the same time, $PM_{2.5}$
concentration started to decrease. Based on the analysis, the pollution ended up at 12:00. The
calculation shows regional transport contributed 62%, relatively lower than that during episode 1 (Fig.

27   6).

Rather than chemical reaction, aged aerosols settled down and had important contribution to the
pollution in episode 2. Vertical observations found that the aged aerosol settled down and caused the
explosive increase of SNA in such a short time, which can't be explained by the ground-level
observations. It was also noticed that the pollution occurred when the emission control plan just
started, which means this episode was partly caused by regional transport before control. Even when




local emission control was conducted effectively, the uncontrolled regional emission still led to severe
pollution in Beijing.
**3.2.3 Pollution process in episode 3**
During episode 3 (November 6th to 11th), Luilihe site recorded a relatively high average $PM_{2.5}$
concentration of $89\pm61\mu g/m^3$. Furthermore, this episode is characterized by much more and faster
increases in OM concentrations than SNA (Fig. 2 and Fig. 3). Specifically, concentrations of aerosol
related with fuel combustion (HOA) increased significantly. While SNA increased slowly ($NH_4^+$ and
$NO_3^-$) or changed little ($SO_4^{2-}$). All of these indicate primary emission rather than the formation of SA
was the dominant cause.
Vertical extinction coefficient shows pollutants appeared at 2000-2500m on November 7th. The air
mass came from the northwest and the vertical convection bringing them down on November 7th and
8th (Fig. 7, Fig. 8 and Fig. 10). Air mass trajectories at 1000 m also show air mass arrived in Beijing
from the south on November 7th but changing to the northwest on November 8th (Fig. S10). Because
the northwest was less polluted and the effective emission control in BJH region during the APEC,
the regional transport of PM was weakened. This is supported by an estimated regional contribution
of 53% to $PM_{2.5}$ in Beijing, much lower than in episode 1 (75%) and episode 2 (63%).
Figure 11 depicts black carbon (BC) concentrations measured by Aethalometer and OM
concentrations measured by ACSM. They tracked each other well during this episode. Concentrations
of BC, a marker of vehicular emission in urban settings, had two peaks every day. One was in the
early morning and another was in the morning rush hour of 9:00am. The first peak might result from
diesel vehicle emissions (Westerdahl, et al., 2009). This is because transportation of goods to Beijing
via heavy-duty diesel vehicles has been permitted at night only, and the number of trucks was large.
When the regional emission control was conducted effectively and air mass was from relatively clean
areas, traffic emissions in and around the city became the dominant source.
**4. Conclusion**
This study indicates that the meteorology condition on the ground sometime couldn't explain the
pollution process, especially the pollutions impacted by transport significantly. Vertical observation
can provide the vertical meteorological and optical profile, which can help identify the regional
transport episodes. Combining the ground-level observation with information from radars, we can
determine the regional transport influence on air quality.



Three episodes of different types under similar ground meteorological condition were discussed in
this study. In episode 1, particle concentration accumulated under the unfavorable meteorological
condition after transport occurred. The transport pollutants brought organic aerosol and SNA
increased under high RH later. In episode 2, pollutants left from episode 1 kept in the boundary layer
in the region. When vertical wind direction changed to downward, the pollutants were settled down.
As a result, OM and SNA increased much explosively. In episode 3, when control plan had been
conducted for several days, SNA and OA concentration increased much less while HOA and
increased significantly. The pollution might be caused by the primary emission from diesel vehicles.
Our research suggests regional transport of air pollutants has significant contribution (up to 70%) to
severe secondary particle pollution, even when local emission was controlled effectively (53%, such
as in APEC summit). Although lots of efforts were paid to air quality management in Beijing, the
equal efforts need to be paid to regional emission to ensure the clean air. What's more, diesel vehicle
emission at night in Beijing might be an important pollution source and needs further investigation.
**5. Acknowledgements**
This work was supported by the MEP's Special Funds for Research on Public Welfare (201409002),
Strategic Priority Research Program of the Chinese Academy of Sciences (XDB05020300), and
National Natural Science Foundation of China (21521064). The authors also appreciate the support
from Collaborative Innovation Center for Regional Environmental Quality.



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





1 **Figures**

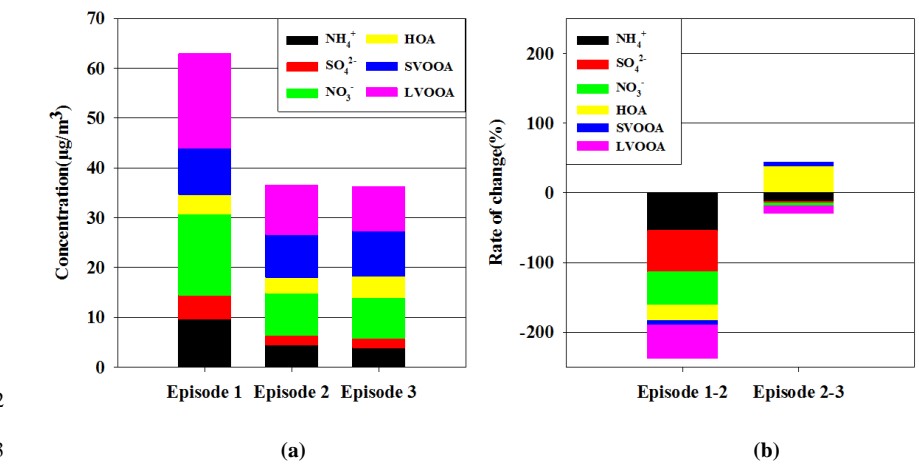

**(a)**                                    **(b)**

**Figure 1. Average PM₁ chemical components and the change rates during different episodes (a)**

**average PM₁ chemical components; (b) change rates in chemical components**

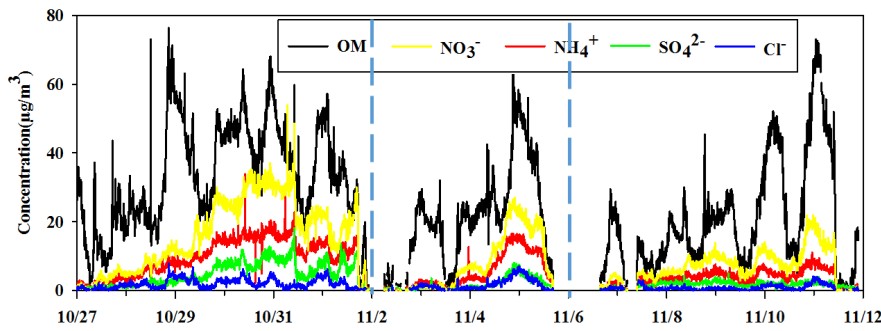

**Figure 2. PM₁ chemical components during the observation at Liulihe site**

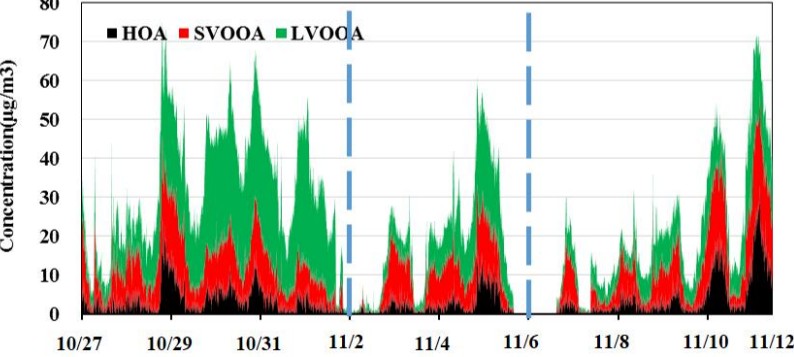

**Figure 3. PM₁ organic components during the observation at Liulihe site**





**Figure 4. Parameters of particles and meteorology during episode 1**

(a) Vertical profile of extinction coefficient (km⁻¹) (Yongledian site); (b) Vertical profile of wind vertical direction and speed (m/s, positive stands for up, negative

stands for down);(c) Horizontal wind direction profile (°, 0° stands for north); (d) wind direction on the ground; (e) Particle size distribution (dN/dlogDp, N:

number concentration (cm⁻³); Dp: particle diameter (nm)); (f) PM₁ chemical components.





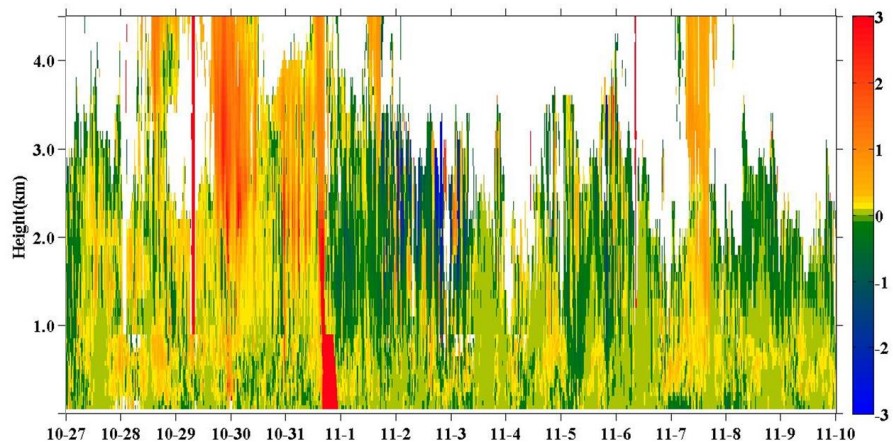

**Figure 5. Vertical profile of wind vertical direction and speed (m/s, positive stands for up and**

**negative stands for down) during the observation time at Liulihe site**





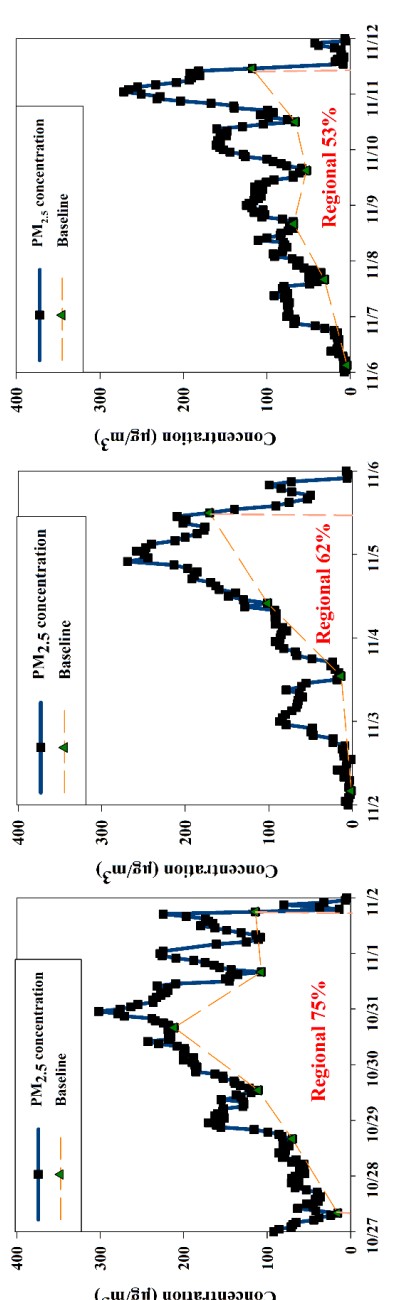

**Figure 6. Regional and local components of the three episodes at Liulihe site**





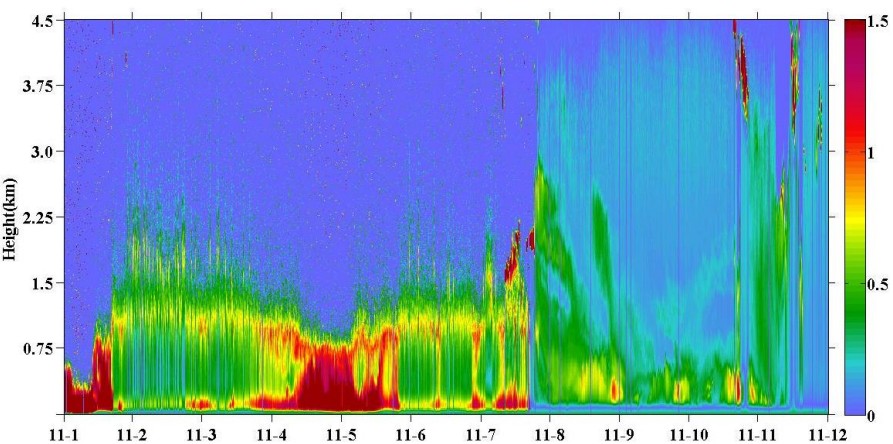

**Figure 7. Vertical profile of extinction coefficient during the observation at Liulihe site**

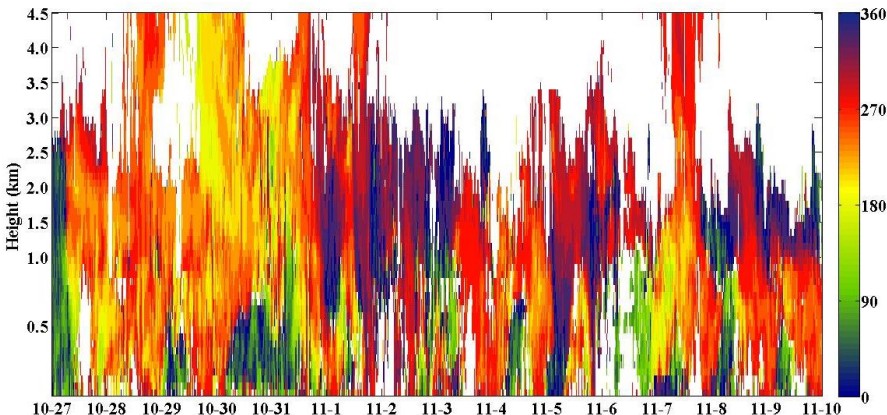

**Figure 8. Vertical profile of wind horizontal direction during the observation at Liulihe site**







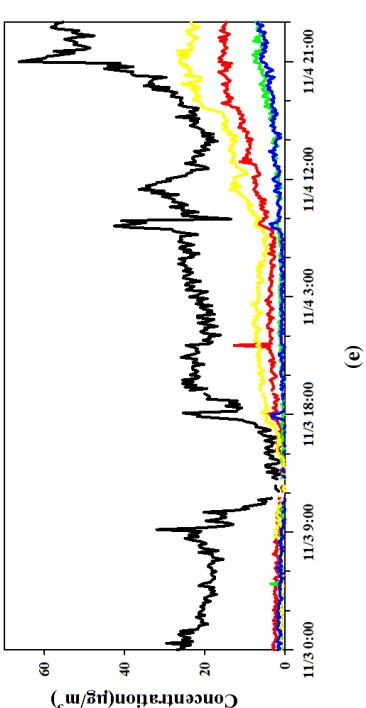

**(e)**

**Figure 9. Parameters of particles and meteorology at Liulihe site during episode 2**

(a) Vertical profile of extinction coefficient (km$^{-1}$); (b) Vertical profile of wind vertical direction and speed (m/s, positive stands for up, negative stands for down); (c) Horizontal wind direction profile (°, 0 ° stands for north); (d) wind direction on the ground; (e) PM$_1$ chemical components.





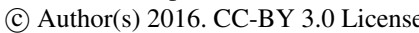

**Figure 10. Parameters of particles and meteorology at Liulihe site during episode 3**

**(a) Vertical profile of extinction coefficient (km⁻¹); (b) Vertical profile of wind vertical direction and speed (m/s, positive stands for up, negative stands for down);(c)**

**Horizontal wind direction profile (°, 0 ° stands for north); (d) wind direction on the ground; (e) PM₁ chemical components.**



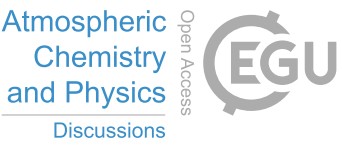

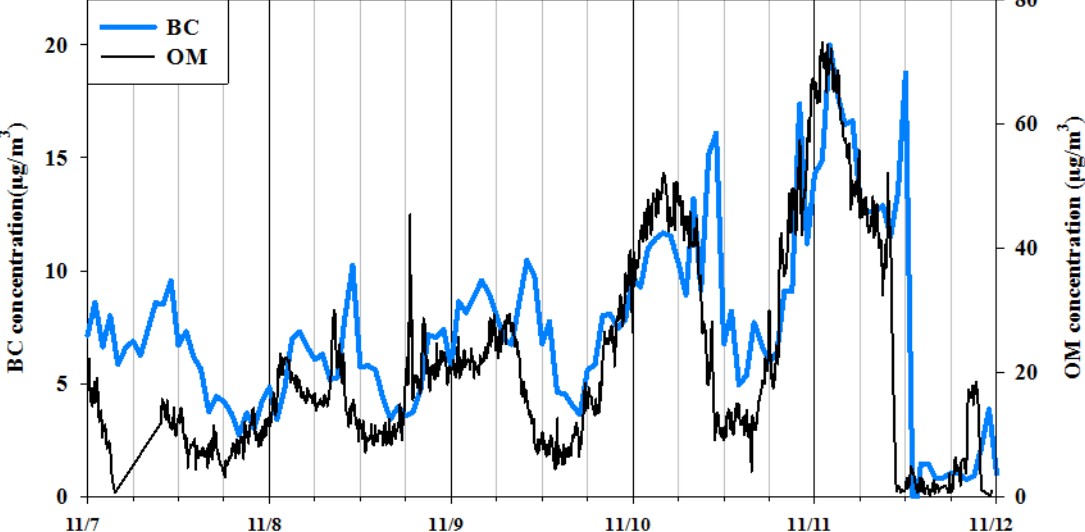

**Figure 11. BC and OM concentrations of PM$_1$ at Liulihe site during episode 3**