# Peer review of "Summit in Beijing"

_Atmospheric Chemistry and Physics, 2016_

## Referee Comment (RC1) · Anonymous Referee #1 · 4 Oct 2016

This manuscript presents observations of air quality and meteorological parameters including ground level data and vertical profiles. Those measurements were used to analyze emissions and atmospheric processes that had significant impacts on PM2.5 air quality during the study period. The topic is relevant to ACP. The approach and applied methods are largely valid. The scientific contribution is good. However, the presentation has room for improvement. My comments and suggestions are listed below.

1. Abstract. The last sentence is unclear, "Further vertical observations are needed to investigate the pollutants transport especially during the explosive increase pollution episode. " 1) if you recommend that others should use vertical methods, it could

read: Future studies may consider including vertical observations to aid investigation of pollutant transport especially during episodic events of rapidly increasing concentrations, or 2) if you believe that what you have done in this study is not enough, it could read: Vertical observations beyond those explored in this study may be necessary to investigate pollutant transport, especially during episodic events of rapidly increasing concentrations.

2. Section 3.1, "Period 1 (October 27th to November 2nd) and period 2 (November 3rd to November 12th) were defined to represent the periods before and during the APEC summit." Given that "Three pollution episodes were selected to discuss the pollution characteristics during the observation (Fig. S3)... Episode 1 (October 27th to November 1st) represents the period before the emission control. Episode 2 (November 2nd to 5th) was the first pollution episode during the emission control plan. Episode 3 (November 6th to 11th) was the second pollution episode during the emission control plan." (pg 7, L10) and "Summit held in Beijing from November 5th to 11th, 2014. A strict air pollution control plan was carried out in the BTH Region to improve air quality in Beijing from November 2nd to 11th for APEC" (pg 4, L3), the selection of Period 1 and Period 2 seem to be rather arbitrary and confusing. I suggest using "Episode 1 and Episodes 2 and 3 combined" and define the three episodes at the beginning of section 3.1.

3. Pg 8, L13, "The high level of PM2.5 is typical in Beijing during the autumn." Please provide the average PM2.5 concentration during this episode.

4. Pg 9, L14, this paragraph seems to be less relevant; it could be better placed in the Supplemental Information.

5. Pg 9, L23," For episode 1, the regional component accounted for 75%", did you mean, "For episode 1, the regional component accounted for 75% of PM2.5 mass concentration observed in Beijing"?

6. Pg 9, L27, "After that vertical wind direction kept downward and promoted the pollutants accumulation, especially SNA." It is uncommon to have a prolonged period and /or a wide spread of downward winds that will result in great changes in atmospheric pressure. The authors may need to provide more data to support this claim or clarify where the downward winds were and for how long.

7. Pg 11, L18, "Concentrations of BC, a marker of vehicular emission in urban settings, had two peaks every day. One was in the early morning and another was in the morning rush hour of 9:00am." The second peak in BC concentrations (blue line in Fig 11) seems to be near noon, please clarify.

8. References were missing at times, e.g. pg 6, L9, HYSPLIT.

9. Figs 9e & 10e, legends seem to be missing.

10. The use of English language is largely satisfactory. However, there are quite a few awkward sentences and word choices, some examples are listed below:

1) Pg 5, L25, it could read, "Vertical wind profiles indicate the transport direction. Vertical RH profiles reflect the strength of heterogeneous reaction at different layers."

2) Pg 7, L30, "10-day observation", the entire observation seems to be either 16 days based on the three episodes (pg 7) or 17 days ("The field campaign was conducted from October 27th to November 12th, 2014." pg 5, L1).

3) Pg 10, L23, it could read: "Dry and clean air mass from the north arrived in. . ."

4) The word "plan" could be omitted in "control plan" after the Introduction.

5) The use of the word "pollution" is unconventional at times, e.g. "It was also noticed that the pollution occurred when the emission control plan just started," (pg 10, L31). Large cities like Beijing would not be free from air pollution, however, the extent of air pollution may vary. Please clarify the meaning of "pollution".

6) Throughout the manuscript, the word "kept" could be replaced by, for example, "continuously", or "was retained".

---

## Referee Comment (RC2) · Anonymous Referee #2 · 7 Oct 2016

The manuscript by Hua et al. investigated the role of regional transport on the formation of PM pollution in Beijing by combining ground and vertical measurements during a unique period (APEC). The results showed that regional transport played a major role in PM pollution but varied substantially among different episodes. In addition, the impact of emission control on PM reduction was also discussed. This study provide new insights into the formation of PM pollution from the view of vertical measurements, and the conclusions showed that vertical measurements are of critical to explain the ground observations. This manuscript falls within the scope of ACP. I recommend it for publication after addressing the following comments.

1. Abstract: it is better to mention the exact location for the ground and vertical measurements.
2. Page 2, line 1: suggest using "aerosol optical properties, winds, relative humidity, and temperature"
3. Page 5, line 19: change "semi volatility" to "semi volatile"
4. Page 6, line 3: no blind area for CFL-03?
5. Page 6, line 15: Cite Jia et al. (2008) where this technique was developed, not in this study. In addition, the approach in Jia et al. (2008) might have large uncertainties in determining the baseline (the lowest points) because of multiple influences from local emissions, regional transport, and secondary processes, I suggest the authors adding several sentences to discuss the uncertainties in quantification of the contributions of regional transport.
6. Page 7, line 19: It is not appropriate to call "$PM_1$ chemical components" because BC was not included. Either use non-refractory $PM_1$ or adding BC in Figure 1.
7. Please mark the three episodes in Figures 2 and 3, or explain the vertical dash lines in the captions.
8. Please try to combine Figures 4, 9 and 10 in one page for easy reading. Also, add the units for the color bars.
9. The colors of chemical species in the figures should be synchronized, e.g., Figure 1 vs. Figure 2, otherwise, it is very confusing.
10. Figure 4: change "wind vertical direction and wind speed" to "wind vertical speed". The wind direction of "up" and "down" was already mentioned in the notes. Same as in Figure 5 and Figure 9.
11. Suggest combining Figure 5 and Figure 8 together for easy reading and comparisons.

---

## Author Response (AR1)

**Anonymous Referee #1**

This manuscript presents observations of air quality and meteorological parameters
including ground level data and vertical profiles. Those measurements were used to
analyze emissions and atmospheric processes that had significant impacts on $PM_{2.5}$ air
quality during the study period. The topic is relevant to ACP. The approach and
applied methods are largely valid. The scientific contribution is good. However, the
presentation has room for improvement. My comments and suggestions are listed
below.

Response: We thank the referee for the comments which help us improve the quality
of our manuscript. We address the reviewer's comments below. The original
comments are in blue and our responses are in black.

1. Abstract. The last sentence is unclear, "Further vertical observations are needed to
investigate the pollutants transport especially during the explosive increase pollution
episode." 1) if you recommend that others should use vertical methods, it could read:
Future studies may consider including vertical observations to aid investigation of
pollutant transport especially during episodic events of rapidly increasing
concentrations, or 2) if you believe that what you have done in this study is not
enough, it could read: Vertical observations beyond those explored in this study may
be necessary to investigate pollutant transport, especially during episodic events of
rapidly increasing concentrations.

Response: This sentence is to recommend that others should use vertical methods. We
have corrected as the reviewer suggested.

2. Section 3.1, "Period 1 (October 27th to November 2nd) and period 2 (November
3rd to November 12th) were defined to represent the periods before and during the
APEC summit." Given that "Three pollution episodes were selected to discuss the
pollution characteristics during the observation (Fig. S3): Episode 1 (October 27th to
November 1st) represents the period before the emission control. Episode 2
(November 2nd to 5th) was the first pollution episode during the emission control
plan. Episode 3 (November 6th to 11th) was the second pollution episode during the
emission control plan." (pg 7, L10) and "Summit held in Beijing from November 5th
to 11th, 2014. A strict air pollution control plan was carried out in the BTH Region to improve air quality in Beijing from November 2nd to 11th for APEC" (pg 4, L3), the selection of Period 1 and Period 2 seem to be rather arbitrary and confusing. I suggest using "Episode 1 and Episodes 2 and 3 combined" and define the three episodes at the beginning of section 3.1.

Response: First, the control plan started from November $3^{rd}$ instead of November $2^{nd}$. We apologize for this typo. To address the comment by the reviewer, we removed the definition of "period 1" and "period 2" from the manuscript and only discuss the characteristics changes before and after control. As suggested by the reviewer, we defined the three episodes at the beginning of section 3.1.

3. Pg 8, L13, "The high level of $PM_{2.5}$ is typical in Beijing during the autumn." Please provide the average $PM_{2.5}$ concentration during this episode.

Response: The average concentration of $PM_{2.5}$ reached to $140 \mu g/m^3$. It has been added in the article as the reviewer suggested.

4. Pg 9, L14, this paragraph seems to be less relevant; it could be better placed in the Supplemental Information.

Response: We agree. It has been placed in the Supplemental Information.

5. Pg 9, L23," For episode 1, the regional component accounted for 75%", did you mean, "For episode 1, the regional component accounted for 75% of $PM_{2.5}$ mass concentration observed in Beijing"?

Response: yes, to be more accurate, we mean the regional component accounted for 75% of $PM_{2.5}$ mass concentration observed at Liulihe site. We have corrected in the article.

6. Pg 9, L27, "After that vertical wind direction kept downward and promoted the pollutants accumulation, especially SNA." It is uncommon to have a prolonged period and/or a wide spread of downward winds that will result in great changes in atmospheric pressure. The authors may need to provide more data to support this claim or clarify where the downward winds were and for how long.

Response: Both the atmosphere pressure and wind speed decreased from October $30^{th}$ to November $1^{st}$ (as shown in Figure R1). This indicates that Liulihe site was probably in the rear of cold anticyclone. The steady weather conditions promoted the accumulation of air pollutants. Meanwhile, the vertical downward wind was unfavorable for pollutants dispersion. Weather Research & Forecasting Model (WRF)

modeling results also show the whole region was under control of weak downward wind from late night on October 30th. (Figure R2, modeling parameters are provided in supplemental information). As a whole, the pollutants were easily accumulated. We agree that "After that vertical wind direction kept downward and promoted the pollutants accumulation, especially SNA" is a little arbitrary. So we have corrected this statement as well as the discussion in page 9, line 7. The atmosphere pressure figure and regional wind vertical speed figure have been added to the supplemental information.

[Figure]

**Figure R1 Meteorology conditions on the ground during the observation at Liulihe site**

[Figure]

**Figure R2 Regional wind vertical speed**

7. Pg 11, L18, "Concentrations of BC, a marker of vehicular emission in urban settings, had two peaks every day. One was in the early morning and another was in the morning rush hour of 9:00am." The second peak in BC concentrations (blue line in Fig 11) seems to be near noon, please clarify.

Response: we check the data and confirm that the second peak is actually at 10:00-11:00am. The second peak might be resulted from vehicles from outside coming into Beijing. Vehicles not registered in Beijing are banned to come into Beijing in the rush hour (7:00 am to 9:00 am), which reduces the morning peaks and smoothes the traffic flow. The vehicles coming into Beijing reach a peak after morning rush hour (http://wenku.baidu.com/link?url=SjtPVT1tgo4ON0KDQ5py8ehw1ZAzUr3k0mSd74 D3F-8lOQZPPvedZiro6E5-MOeFFuww7VZjy3XwRqfU-mHXkg0_8kSy5p9FGyokfr FZX0e). As a result, a second peak appeared in the late morning at Liulihe site where is close the entrance from Hebei Province into Beijing. We have corrected the discussion as suggested.

8. References were missing at times, e.g. pg 6, L9, HYSPLIT.

Response: We check all the references in the article and corrected/added the following references.

Add missing reference: Tang, G., Zhu, X., Hu, B., Xin, J., Wang, L., Münkel, C., Mao, G. and Wang, Y. (2015) Impact of emission controls on air quality in Beijing during APEC 2014: lidar ceilometer observations. Atmospheric Chemistry and Physics 15(21), 12667-12680.

Adjust reference order: Ng, N.L., Herndon, S.C., Trimborn, A., Canagaratna, M.R., Croteau, P.L., Onasch, T.B., Sueper, D., Worsnop, D.R., Zhang, Q., Sun, Y.L. and Jayne, J.T. (2011) An Aerosol Chemical Speciation Monitor (ACSM) for Routine Monitoring of the Composition and Mass Concentrations of Ambient Aerosol. Aerosol Science and Technology 45(7), 780-794.

Correct the author name of citing reference (page 6, line 1) from "Fernald, 1984" to "Frederick, 1984".

The reference for HYSPLIT and Trajstat is "Wang, Y., Zhang, X. and Draxler, R.R. (2009) TrajStat: GIS-based software that uses various trajectory statistical analysis methods to identify potential sources from long-term air pollution measurement data. Environmental Modelling & Software 24(8), 938-939."

9. Figs 9e & 10e, legends seem to be missing.

Response: we have added the legends for the figures.

10. The use of English language is largely satisfactory. However, there are quite a few awkward sentences and word choices, some examples are listed below:

Response: We have carefully checked the article and polished the sentences. In addition, a copy-editing team will further help to improve the language after the manuscript is accepted for publication at ACP.

1) Pg 5, L25, it could read, "Vertical wind profiles indicate the transport direction. Vertical RH profiles reflect the strength of heterogeneous reaction at different layers."

Response: we have corrected as the suggested.

2) Pg 7, L30, "10-day observation", the entire observation seems to be either 16 days based on the three episodes (pg 7) or 17 days ("The field campaign was conducted from October 27th to November 12th, 2014." pg 5, L1).

Response: we have corrected the description of observation days.

3) Pg 10, L23, it could read: "Dry and clean air mass from the north arrived in…"

1 Response: we have corrected as the suggestions.

2 4) The word "plan" could be omitted in "control plan" after the Introduction.

3 Response: we have corrected as the suggestions.

4 5) The use of the word "pollution" is unconventional at times, e.g. "It was also
5 noticed that the pollution occurred when the emission control plan just started," (pg
6 10, L31).

7 Large cities like Beijing would not be free from air pollution, however, the extent of
8 air pollution may vary. Please clarify the meaning of "pollution".

9 Response: we check the article and clarify all the pollution might not be clear.

- Correct "large quantity of emissions has caused serious air pollution in China" (page 3, line 3) to "large quantity of emissions has caused serious particulate matter pollution in China."
- Correct "The significantly reduced local emissions led to reduced complexity of pollution process" (page 4, line 15) to "The significantly reduced local emissions led to reduced complexity of particulate matter pollution process".
- Correct "However, the general characteristics derived from ground-level observation are insufficient to identify the leading cause of air pollution, local emissions, regional transport, or both." (page 8, line 4) to "However, the general characteristics derived from ground-level observation are insufficient to identify the leading cause of particulate matter pollution, local emissions, regional transport, or both.".
- Replace the word "pollution" in the sentence "Rather than chemical reaction, aged aerosols settled down and had important contribution to the pollution in episode 2." (page 10, line 24) to "high $PM_{2.5}$ concentration".
- Correct the sentence "Even when local emission control was conducted effectively, the uncontrolled regional emission still led to severe pollution in Beijing." (page 10, line 29) to "Even when local emission control was conducted effectively, the uncontrolled regional emission still led to severe particulate matter pollution in Beijing.".
- Correct the sentence "This study indicates that the meteorology condition on the ground sometime couldn't explain the pollution process, especially the pollutions impacted by transport significantly." (page 11, line 27) to "This study indicates that the meteorology condition on the ground sometime couldn't explain the air pollution process, especially the air pollution episodes significantly impacted by regional transport of air pollutants".

6) Throughout the manuscript, the word "kept" could be replaced by, for example, "continuously", or "was retained".

Response: we check the word in the article and replace some of them.

- We replace the "kept" in the sentence "it still kept in the southwest above 500m, indicating significant influence of regional transport." (page 9, line 5) to "was retained".
- We replace the "kept" in the sentence "RH was high, wind speed kept low and wind direction was dominated by southwest in the surface." (page 9, line 22) to "was continuously".
- We replace the "kept" in the sentence "In episode 2, pollutants left from episode 1 kept in the boundary layer in the region." (page 12, line 4) to "was retained".

The manuscript by Hua et al. investigated the role of regional transport on the formation of PM pollution in Beijing by combining ground and vertical measurements during a unique period (APEC). The results showed that regional transport played a major role in PM pollution but varied substantially among different episodes. In addition, the impact of emission control on PM reduction was also discussed. This study provide new insights into the formation of PM pollution from the view of vertical measurements, and the conclusions showed that vertical measurements are of critical to explain the ground observations. This manuscript falls within the scope of ACP. I recommend it for publication after addressing the following comments.

We thank the referee for supporting the publication. We will put more efforts to improve the quality of our manuscript. We address the reviewer's comments below. The original comments are in blue and our responses are in black.

1. Abstract: it is better to mention the exact location for the ground and vertical measurements.

Response: the location has been added in the abstract.

2. Page 2, line 1: suggest using "aerosol optical properties, winds, relative humidity, and temperature"

Response: we have corrected as suggested.

3. Page 5, line 19: change "semi volatility" to "semi volatile"

Response: we have corrected as suggested.

4. Page 6, line 3: no blind area for CFL-03?

Response: There are 300m blind area for CFL-03. We have added it to the article.

5. Page 6, line 15: Cite Jia et al. (2008) where this technique was developed, not in this study. In addition, the approach in Jia et al. (2008) might have large uncertainties in determining the baseline (the lowest points) because of multiple influences from local emissions, regional transport, and secondary processes, I suggest the authors adding several sentences to discuss the uncertainties in quantification of the contributions of regional transport.

Response: we had cited in page 6, line 19 in the original manuscript. We have cited in line 15 as suggested.

The uncertainties discussion is also added to the manuscript. The uncertainty evaluation mainly includes systematic errors, random errors and sensitivities. The major systematic errors depend on the calibration of instruments for $PM_{2.5}$ concentration measurement. Minor systematic errors might be from the judging the location and height of the daily minima and the sensitivities analysis suggests these errors are less than 10%. Random errors include data measurement and quantification step, such as identifying the daily minima properly, dealing with days without less-obvious afternoon minima and using linear interpolation between the daily minima. All these errors are evaluate by Jia et al. (2008). As a whole, this technique has an uncertainty of 40%-50% for results of daily regional transport.

6. Page 7, line 19: It is not appropriate to call "PM1 chemical components" because BC was not included. Either use non-refractory PM1 or adding BC in Figure 1.

Response: we correct the "$PM_1$ chemical components" to "non-refractory $PM_1$ chemical components".

7. Please mark the three episodes in Figures 2 and 3, or explain the vertical dash lines in the captions.

Response: we have corrected the figures as suggested.

8. Please try to combine Figures 4, 9 and 10 in one page for easy reading. Also, add the units for the color bars.

Response: we have corrected the figures as suggested.

9. The colors of chemical species in the figures should be synchronized, e.g., Figure 1 vs. Figure 2, otherwise, it is very confusing.

Response: we have corrected the figures as suggested.

10. Figure 4: change "wind vertical direction and wind speed" to "wind vertical speed". The wind direction of "up" and "down" was already mentioned in the notes. Same as in Figure 5 and Figure 9.

Response: we have corrected the figures as suggested.

11. Suggest combining Figure 5 and Figure 8 together for easy reading and comparisons.

Response: we have corrected the figures as suggested.

**MANUSCRIPT**

[revised manuscript text omitted]

**S4 Weather Research & Forecasting Model (WRF) modeling analysis**

WRF version 3.7 is utilized to generate the regional meteorological fields. The parameters have been introduced in our previous studies (Wang et al., 2015).

**Table S1. Instruments information at Liulihe site.**

| Measurement index | Instruments | Time resolution |
|---|---|---|
| $PM_{2.5}/PM_{10}$ | TEOM1405/1400a (Thermo Scientific, USA) | 1hour |
| $SO_2$ | API100E (Teledyne, USA) | 1hour |
| $NO_2$ | API200E (Teledyne, USA) | 1hour |
| $O_3$ | API400E (Teledyne, USA) | 1hour |
| Off-line $PM_{2.5}$ | Partisol 2300 (Thermo Scientific, USA) | 23.5 hour |
| NR-$PM_1$ chemical composition ( $SO_4^{2-}$, $NO_3^-$, $NH_4^+$, $Cl^-$, Organic Matter) | ACSM (Aerodyne Research Inc. USA) | 8min |
| Particle size distribution | Nano SMPS&SMPS&APS 3321 (TSI Inc, USA) | 5min |
| Absorption coefficient/black carbon | Aethalometer AE42 (Margee Scientific, USA) | 1 min |
| Meteorological data (RH, wind speed/direction, temperature, atmospheric press) | WXT520 (VAISALA, Finland) | 1hour |
| Wind profile | CFL-03 (23rd Institute of China Aerospace Science and Industry Corporation) | 6min |
| Temperature and humidity profile | QFW-6000 (22nd Institute of China Electronic Technology Group Corporation) | 2min |

[Figure]

**Figure S1. Field observation site location**

[Figure]

**Figure S2. Correlation between NR-PM$_1$ (= Organic matter + SO$_4^{2-}$ + NO$_3^-$ + NH$_4^+$ + Cl$^-$) measured by the ACSM and PM$_{2.5}$ by the TEOM**

[Figure]

**Figure S3. Factor profile performed by PMF**

[Figure]

(a)                                                                              (b)

**Figure S4. Average concentration and change rate of pollutants during the observation. (a) Average concentration of pollutants; (b) Change rate of pollutants**

[Figure]

**Figure S5. Hourly PM₂.₅ concentrations at Liulihe and Miyun during the observation**

[Figure]

[Figure]

**Figure S6. Meteorology conditions on the ground during the observation at Liulihe site**

[Figure]

**F**igure S7. Air mass trajectory analysis during episode 1

[Figure]

**Figure S8. Regional wind vertical speed generated by WRF**

[Figure]

**Figure S9. Vertical profile of extinction coefficient at Baoding site during episode 2 (km⁻¹)**

[Figure]

**Figure S10. Air mass trajectory analysis during episode 2**

[Figure]

**Figure S11. Air mass trajectory analysis during episode 3**